# Investigating Feature Alignment Under An Infant-Inspired Visual Distribution Shift

## Abstract

Recent work on visual learning in people finds that human infants often experience extended bouts of experience with a small number of familiar objects (e.g., toy ducks at home), with a very long tail of less frequent exposures to less familiar objects (e.g., real ducks at the park). When facing this type of distribution shift between toy ducks and real ducks, learners trying to build coherent representations that bridge these two distributions can leverage at least two distinct types of learning signals: (1) categorical learning signals, which explicitly assign two inputs to the same class (e.g., hearing adults label both toy ducks and real ducks with the same word, "duck;" and (2) perceptual learning signals, which implicitly assign two inputs to the same class because of perceived similarities (e.g., both toy ducks and real ducks have bills, wings, and webbed feet). In this paper, we examine how these two types of learning signals interact to impact a learner's cross-domain classification performance, through the lens of feature alignment as an interim goal for the learner. We propose new cluster-based metrics to quantify feature alignment in an infant-inspired two-domain learning problem, and we describe a series of experiments that systematically vary these learning signals to observe impacts on feature alignment and overall learning outcomes.

## 1 Introduction

Human infants are powerful learners. A slew of recent work in developmental psychology finds that the distributions of learning experiences that infants receive in their daily lives are often highly non-uniform in content, space, and time, which raises interesting questions about how robust learning can arise from such distributions. In the case of visual learning about object categories, one salient distributional property is that infants often experience extended bouts of experience with a small number of familiar objects (e.g., toy ducks at home) Herzberg et al. (2022), with a very long tail of less frequent exposures to less familiar objects (e.g., real ducks at the park).

This particular pattern is even more interesting because of the substantial visual differences in the two types of experiences—a distribution shift between toy ducks and real ducks. Learners trying to build coherent representations that bridge these two distributions can leverage at least two distinct types of learning signals: (1) *Categorical* learning signals, which explicitly assign two inputs to the same class (e.g., hearing adults label both toy ducks and real ducks with the same word, "duck;" and (2) *Perceptual* learning signals, which implicitly assign two inputs to the same class because of perceived similarities (e.g., both toy ducks and real ducks have bills, wings, and webbed feet).

In ML research on distributions shifts, e.g., in domain adaptation, it is often assumed that a learner can succeed in bridging two domains by developing features that are aligned across domains. Both categorical learning signals and perceptual learning signals can contribute to feature alignment, but it is not always obvious how the two signals contribute, and especially how they might interact to strengthen or weaken learning outcomes. In this paper, we aim to disentangle and characterize these learning processes in the context of an infant-inspired distribution shift. Our contributions are:

- We propose new cluster-based metrics to quantify feature alignment across category and domains in a two-domain learning problem, particularly to look at per-category density as well as per-category and full-domain overlap.

- Through our first set of experiments, we show that varying the perceptual learning signal does not influence the network's ability to effectively classify unseen images from both datasets. In fact, the performance remains comparable for many different configurations of the feature space.
- In our second experiments, we show that there is a key difference in the way the network treats inconsistent perceptual vs category signals. While the network learns to apply the inconsistent labelings to unseen images, the network ignores inconsistent perceptual labelings.
- Third, we show that the network learns some form of relationship between corresponding classes from the two datasets even when no such information has been presented to the network.

## 2 INFANT-INSPIRED DISTRIBUTION SHIFT

Humans are experts at recognizing and categorizing thousands of object classes very effectively and robustly. Developmental psychology research have suggested that the visual experience of infants plays a crucial role for learning to distinguish between visual categories Yurovsky et al. (2013). Specifically, these studies suggest that the visual experience infants get when playing with objects provides the foundation on which robust object recognition abilities are developed. A recent line of research has looked at identifying key properties of an infant's visual experience when they are playing with objects — they find that *repeated* experiences with a small number of *individual* objects forms a major part of their visual experience Herzberg et al. (2022). Further research suggests that these patterns of visual experience with individual objects during playing is seen in subsistence communities as well Casey et al. (2022).

When playing with their toys and other household objects, infants gain a lot of experience with these individual objects. They see each object from a large number of viewpoints and under many kinds of varying conditions. In contrast, adult visual experience of the world is filled with numerous objects, but the range of viewpoints from which each object is seen is very limited. On the other hand, infants play with a relatively small number of objects, while the outside world is filled with different kinds of object instances. In machine learning terms, this can be characterized as a *distribution shift* — while the infant's visual experience is viewpoint-rich, but limited in instances, adult visual experience is more limited in viewpoints, but we see a much larger number of instances.

This type of distribution shift is highly relevant not just for understanding human learning but also for the study of any learning agent that is *embodied*. The reason that infants receive an instance-limited and viewpoint-rich distribution of visual experience is due to the constraints and affordances of having a physical body. While only a small number of objects are within reach (relative to all that exist in the world), an embodied agent can get *lots* of experience with each of those nearby objects.

### 2.1 DATASETS

We now describe the datasets we use to operationalize the distribution shift. To represent the characteristics of infant visual experience, we use the Toybox dataset (Wang et al., 2018). To represent the instance-rich visual experience of the world, we curate a dataset using images from the ImageNet (Deng et al., 2009) and the MS-COCO (Lin et al., 2014) datasets. Specifically, we extract a set of images from these datasets corresponding to the classes in the Toybox dataset. We call this the IN-12 dataset.

**Toybox dataset** The Toybox dataset contains short egocentric videos of objects being manipulated in different ways. The dataset contains 360 objects from 12 categories; each category is among the early learned nouns among children in the US (Fenson et al., 2007). The categories in the dataset can be grouped into 3 super-categories: vehicles (airplanes, cars, helicopters, trucks), animals (cat, duck, giraffe, horse) and household objects (balls, cups, mugs, spoons).

We use the Toybox dataset in our experiments for the following reasons: (1) The fact that the object categories correspond to the early learned nouns in children increases the developmental relevance of the object set considered. (2) The videos in the dataset are captured using head-mounted cameras. Additionally, the videos capture different kinds of object manipulations, such as rotation and random manipulation (called *hodgepodge* in the dataset). Thus, for each object, there exists a wide range of viewpoints from which that object is seen.

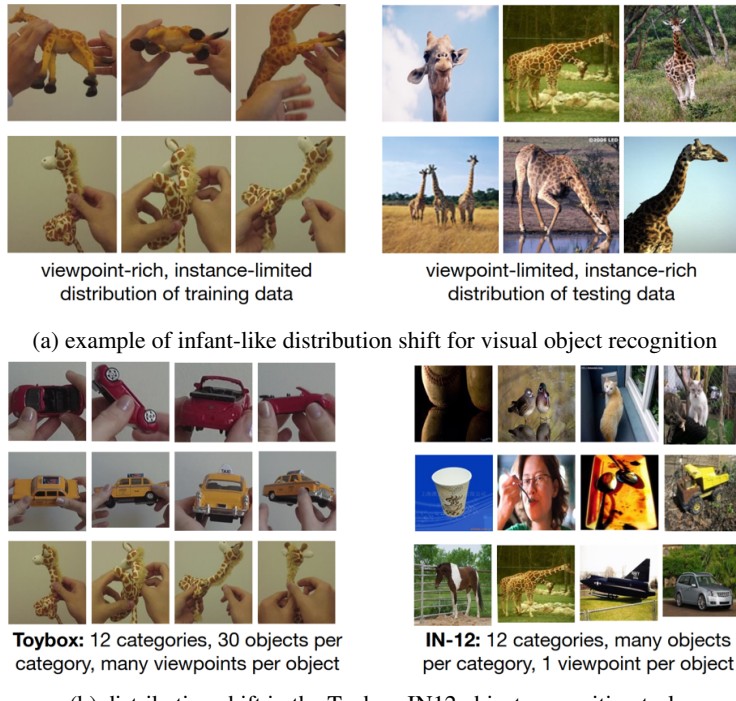

(a) example of infant-like distribution shift for visual object recognition

(b) distribution shift in the Toybox-IN12 object recognition task

Figure 1: The Toybox → IN-12 distribution shift problem. The distribution shift mimics the distribution shift encountered in an infant's visual experience.

**IN-12 dataset** We have curated the IN-12 dataset from the ImageNet (Deng et al., 2009) and the MS-COCO (Lin et al., 2014) datasets. Our target images are extracted from the ImageNet and MS-COCO datasets for the 12 Toybox classes. To do this, we first manually extract all ImageNet classes which correspond to the 12 categories in Toybox. From among these extracted candidate classes, we select a few synsets which describe the category at a general level. For example, we select the 'car' class and disregard the 'police car' category. Then, from these chosen synsets, we extract 1600 images per class to form our target dataset. While selecting the images, we ensured that each candidate synset was equally represented among the selected images. We have provided the list of the selected ImageNet synsets for each Toybox class in the appendix. However, we found that the chosen synsets for giraffes and helicopters did not have enough images, so for these classes, we added images from the MS-COCO dataset. This combined dataset containing images from ImageNet and MS-COCO for the Toybox classes, which we call the IN-12 dataset, serves as our target dataset.

## 2.2 COMPARISON WITH OTHER DISTRIBUTION-SHIFT DATASETS

Despite the success of deep learning methods at several computer vision tasks, these methods still remain susceptible to distribution shift, i.e. they fail to generalize when the test distribution differs from the training distribution (Torralba & Efros, 2011). Thus, addressing the distribution shift problem has received a lot of focus in recent years. Several visual tasks exist which look at handling different kinds of distribution shift; within the image classification paradigm, tasks like domain adaptation (Ben-David et al., 2010) and domain generalization (Blanchard et al., 2011) look at bridging this problem in different ways.

Several datasets have been proposed that consider different kinds of distribution shift. Datasets such as Office-31 (Saenko et al., 2010), Office-Caltech (Gong et al., 2012), PACS (Li et al., 2017), Office-Home (Venkateswara et al., 2017), VisDA-2017c (Peng et al., 2017), DomainNet (Peng et al., 2019), ImageNet-R (Hendrycks et al., 2021)are some of the datasets commonly used in distribution shift problems. However, the distribution shift we consider in this paper is qualitatively very different from those considered previously. For example, the Office-31 dataset handles differences in camera

and image source. Similarly, datasets like PACS and DomainNet consider changes in the rendition styles of the images (clipart, catoon, painting, etc.). However, these are quite different from the distribution shift we are considering in this paper. The distribution shift in the VisDA-2017c dataset is closer to what we are considering in this paper; however, a key distinction is that the source dataset in the VisDA-2017c dataset contains 2d renderings of simple 3d models. As such, they do not capture the rich variety of texture and color as seen during an infant's visual experience.

## 3 OUR APPROACH

In our experiments, we want to investigate the interactions between categorical and perceptual learning signals in shaping the feature space and enabling effective classification of images belonging to both datasets. We do this by training convolutional neural networks (CNNs) (LeCun et al., 1998) jointly using images from both datasets. Through our experiments, we systematically vary the nature of the category and perceptual signals used to train the network. Category signals are presented as ground truth labels to the network and as such are used to compute the cross-entropy loss for the training images. In different experiments, we vary whether the category signal given to images from the two datasets are consistent or not. Specifically, the different settings encode answers to the following question: *given a Toybox car image and an IN-12 car image, does the ground truth knowledge presented to the network classify both images as cars?* In the **Consistent** setting, the answer is yes. In the **Inconsistent** setting, the answer is no — in this case, the labels for the IN-12 images are shuffled so all IN-12 cars would be labeled the same as some other Toybox class, say giraffes. In the **Different** setting, Toybox and IN-12 cars have distinct labels attached to them and IN-12 images are not mislabeled. In the **Agnostic** setting, the network has no knowledge about the relative correspondences of the Toybox and IN-12 images.

The perceptual signals, on the other hand express equivalences between groups of images from the two datasets. During training, we utilize the perceptual signal as a means of encouraging alignment between two sets of images, one from the Toybox dataset and the other from the IN-12 dataset. In this case, the different settings encode answers to the question: *over which sets of images from the two datasets does the equivalence exist?* In the **None** setting, no equivalence is encouraged between any sets of images between the two datasets. In the **Global** setting, the network is encouraged to align the feature-space representations of the Toybox images in a minibatch with the representations of the IN-12 images in the minibatch. In the **Class** setting, the network tries to align images from the corresponding classes together in the feature space. The exact form of the alignment signals are discussed in Sections 4 and 5. Finally, in the **Diverged** setting, the network is explicitly encouraged to separate the two datasets in the feature space. Fig 2 provides a comprehensive overview of the variations in the learning signal that we use in our experiments and which experiments contain which variations. More details about the exact form of both the category learning signal and the perceptual learning signal are presented in the relevant sections later. Note that in the (*) case shown in the table, the learning signal explicitly tries to align different categories from the two datasets in the feature space.

| | Setting | Category Signal | | | | Perceptual (Alignment) Signal | | | |
|---|---|---|---|---|---|---|---|---|---|
| | | Consistent | Different | Inconsistent | Agnostic | None | Global | Class | Diverged |
| Experiment 1 | TB + IN-12 | ✓ | | | | ✓ | | | |
| | TB + IN-12 + Aligned | ✓ | | | | | ✓ | | |
| | TB + IN-12 + Class Aligned | ✓ | | | | | | ✓ | |
| | TB + IN-12 + Diverged | ✓ | | | | | | | ✓ |
| | TB + IN-12 + Class Diverged | | ✓ | | | | | | ✓ |
| Experiment 2 | TB + IN-12 Scrambled | | | ✓ | | ✓ | | | |
| | TB + IN-12 Scrambled + Global Aligned | | | ✓ | | | ✓ | | |
| | TB + IN-12 Scrambled + Class Aligned | | | ✓ | | | | ✓* | |
| Experiment 3 | TB + IN-12 2FC | | | | ✓ | ✓ | | | |
| | TB + IN-12 2FC + Aligned | | | | ✓ | | ✓ | | |

Figure 2: We train our model jointly using images from the Toybox and IN-12 datasets. Two kinds of learning signals are presented to the model during training: category signals and perceptual signals. This table shows the different variants of both signals used in our experiments as well as which experiments they are used in.

### 3.1 RECOGNIZING PATTERNS IN THE FEATURE SPACE

We are interested in investigating the local structure that develops in the feature space of the network during training. To do this, we use the UMAP technique (McInnes et al., 2018) to reduce the dimensionality of the feature space. UMAP is a manifold learning technique that tries to find low-dimensional embeddings for each data point while maintaining the local structure of the dataset as seen in the high-dimensional space. UMAP is commonly used to visualize the local structure in high-dimensional data in several scientific applications (Sun et al., 2022). Next, we use the UMAP embeddings to calculate different metrics which capture different properties of the feature space.

**Preprocessing**  Before computing the metrics, we first normalize the UMAP embeddings to fit them in the range [-1, 1]. The purpose of this step is to make the calculated metrics from UMAP embeddings comparable across different models.

**Outlier Removal**  The calculated metrics are often sensitive to the presence of outliers in the dataset, thus necessitating the detection and removal of these outliers. We want to clarify here that the outlier detection steps are done for every category for each dataset separately. For this, after normalizing the embeddings, we build a minimum spanning tree for the relevant datapoints. Subsequently, we remove some of the longer edges of this MST, thus leading to the formation of several disconnected components. We then select the points in the largest of these connected components as the core cluster points for which the metrics are calculated. We apply an adaptive threshold to select which edges will be dropped from the minimum spanning tree. Specifically, we calculate $q_{2.5}$ and $q_{97.5}$ and set the threshold as $\text{thresh} = q_{97.5} + 1.5 * (q_{97.5} - q_{2.5})$. Any edge that has a weight greater than this threshold is dropped from the MST. This threshold ensures that less than $5\%$ of the edges are dropped.

**Measuring density**  Given the core datapoints for a particular class and dataset, we define the density of that embedding set as the quotient between the number of corepoints remaining after pruning and the area of the convex hull for the corepoints.

**Measuring overlap**  We use kernel density estimation to measure the overlap between two sets of datapoints. Specifically, given two sets of datapoints $C_1 = \{x_1, x_2, \ldots, x_n\}$ and $C_2 = \{y_1, y_2, \ldots, y_n\}$, we first fit a kernel density estimator to $C_1$, $f_1(x)$ given by:

$$f_1(x) = \frac{1}{n} \sum_{i=1}^{n} K(\frac{x - x_i}{h})$$

where K is the kernel and $h$ is the width of the kernel, known as the bandwidth. Finally, given the learned density model, we then define the log likelihood of points $C_2$ under the density model $f_1(x)$ as $D_{2|1} = \prod_{i=1}^{n} p(y_i|f_1)$. Similarly, we construct another measure $D_{1|2}$ in the reverse direction. We then compute the mean of these two values and use this as a measure for the overlap between the two clusters. In our experiments, we have used the Gaussian kernel to compute the kernel densities. The kernel density estimation method is sensitive to the bandwidth parameter. We performed a 5-fold cross-validation on the training set to select the optimal bandwidth value.

## 4 EXPERIMENT 1

In Experiment 1, we fix the category learning signal — the network has access to the correct class labels for images from both datasets. We use a joint supervised training framework: we train a CNN using a supervisory signal that evaluates the network's ability to classify training images from both datasets. We note here that during the forward propagation through the network, each mini-batch contains equal number of images from each dataset. We empirically found that this approach generalizes better than separate forward propagations for each dataset.

In addition to the category signal, we use additional perceptual signals to encourage or discourage alignment between the features in the representation space. We systematically vary the perceptual signal to encourage different kinds of organization of the feature space. The variations on the learning signal we have are described below:

- TB + IN-12: This setting is the baseline setting where we do not use any kind of additional learning signal other than the joint supervisory signal. Specifically, the loss function used to train the network is given by $\mathcal{L}_{joint} = l_{ce\_tb} + l_{ce\_in12}$, where $l_{ce\_tb}$ is the cross-entropy loss for the Toybox images and $l_{ce\_in12}$ is the cross-entropy loss for the IN-12 images in a minibatch.

- TB + IN-12 + Aligned: In this setting, we encourage the CNN to learn representations where the representations for images from the Toybox and IN-12 dataset are aligned. We do this by computing a discrepancy metric between images from the two datasets and minimizing this metric in addition to $l_j$. Specifically, we use the maximum mean discrepancy (mmd) metric (Gretton et al., 2012) between the feature space representation of images from the two domains. The mmd-loss function we use in our experiments is given by

$$l_{mmd} = \frac{1}{n_s^2} \sum_{x \in X_s} \sum_{y \in X_s} k(x,y) + \frac{1}{n_t^2} \sum_{x \in X_t} \sum_{y \in X_t} k(x,y) - \frac{2}{n_s n_t} \sum_{x \in X_s} \sum_{x \in X_t} k(x,y)$$

  where $X_s$ are the set of images from Toybox and $X_t$ are the set of images from IN-12 in a mini-batch and $n_s = |X_s|, n_t = |X_t|$. We then minimize a combined loss $\mathcal{L} = \mathcal{L}_{joint} + \gamma l_{mmd}$ to train the network.

- TB + IN-12 + Class-aligned: In this setting, we modify the mmd loss function to encourage explicit class alignment between corresponding classes from the two domains. The specific form of the loss function is given by:

$$l_{mmd}^c = \frac{1}{n_{sc}^2} \sum_{x \in X_{sc}} \sum_{y \in X_{sc}} k(x,y) + \frac{1}{n_{tc}^2} \sum_{x \in X_{tc}} \sum_{y \in X_{tc}} k(x,y) - \frac{2}{n_{sc} n_{tc}} \sum_{x \in X_{sc}} \sum_{x \in X_{tc}} k(x,y)$$

  where $X_{sc} = \{x_i \in X_s | y_i = c\}, X_{tc} = \{x_i \in X_t | y_i = c\}, n_{sc} = |X_{sc}|$ and $n_{tc} = |X_{tc}|$. The combined loss function used to train the network is given by $\mathcal{L} = \mathcal{L}_{joint} + \gamma \ell_{ccmmd}$, where $\ell_{ccmmd} = \sum_{c \in \mathcal{C}} l_{mmd}^c$.

- TB + IN-12 + Diverged: Here, we put pressure on the network to learn representations so it is possible to distinguish between images from the two datasets in the feature space using a multi-layer perceptron. We used a separate classification head that predicts which dataset each image belongs to using a multi-task learning approach. Given an image-source pair $(x,y)$, where $y \in \{$Toybox, IN-12$\}$, the classification head produces an output $y'$. During training, in addition to the classification loss, we also compute the loss $\ell_{domain} = \sum_{(x,y) \in D} -y \log(y') - (1-y) \log(1-y')$. The combined training loss thus is given by $\mathcal{L} = \mathcal{L}_{joint} + \ell_{domain}$.

- TB + IN-12 + Class-Diverged: In this setting, we want to encourage the feature vectors for each class-domain pair to be distinguishable in the feature space. This is relatively simpler to do than the previous settings. We do this by considering a 24-way classification task, where each output node corresponds to a specific category-domain pair.

## 4.1 Architecture and Experimental Setup

We use a ResNet-18 He et al. (2016) backbone in our experiments. We initialize the network with the Glorot initialization (Glorot & Bengio, 2010) and train the networks from scratch on each of the different experimental settings. We use the Adam optimizer (Kingma & Ba, 2014) for training the network. In every setting, we train the networks for 100 epochs with 500 minibatches per epoch. During training, we linearly increase the learning rate for the first 2 epochs of training and then decay the learning rate using a cosine decay schedule (Loshchilov & Hutter, 2016) without any restarts.

Other than these default settings, we needed some modifications for the TB+IN-12+Class-Diverged and the TB+IN-12+Diverged setting. In the TB+IN-12+Diverged setting, we add a separate MLP with 1000 hidden neurons to correctly recognize which dataset each image belongs to. The loss function in this case is a linear sum of the cross-entropy losses and the logistic loss for identifying the domain. For the TB+IN-12+Class-Diverged setting, we have 24 output classes instead of 12, so the dimensions of the last linear layer were changed. [1]

---

[1] Our code can be accessed at anonymized gdrive link.

## 4.2 RESULTS

Fig 3 shows our results for these set of experiments. In addition to the experiments in the joint supervised setting, we also report accuracy and metric values for the models which were trained separately on the Toybox and IN-12 datasets. We note that all the joint supervised models outperform the model trained solely on the Toybox dataset.

Looking at the joint supervised experiments, we see that all the different models achieve comparable accuracy on both the datasets. This shows that the CNN models are powerful enough to organize the feature space in different ways without any major drop in the accuracy. Secondly, we see that for all the jointly trained models, the average cluster density for the IN-12 images is much higher than that for the Toybox images. Thirdly, we can look at the average overlap between the classes to gauge how well images from different domains overlap in the feature space. For example, the models (TB + IN-12 + Diverged, TB + IN-12 + Class-Diverged) for which we encouraged the network to push the two datasets apart show very large negative values for the metric. In contrast, the models where we encourage alignment between the two domains show large positive values, showing that the features for clusters for different classes effectively overlap in these cases. Similarly, the networks that were trained individually on the two datasets show values in between these two extremes; this is likely due to the distribution gap between the two datasets. It is a bit surprising that the TB + IN-12 model shows such a low value for the overlap measure despite the presence of consistent category signals for both datasets. The network seems to leverage the similarity between images from each dataset to group them separately.

| Setting | Accuracy | | Avg. Density of Cluster | | Avg. overlap b/w domains |
|---|---|---|---|---|---|
| | Toybox | IN-12 | Toybox | IN-12 | |
| Toybox | 76.39 | 24.92 | 125131 | 12571 | -4144 |
| IN-12 | 35.69 | 87.92 | 11123 | 182362 | -2837 |
| TB + IN-12 | 78.86 | 87.08 | 126871 | 198736 | -25866 |
| TB + IN-12 + Aligned | 82.56 | 87.25 | 93598 | 127329 | 3053 |
| TB + IN-12 + Class Aligned | 79.69 | 86.83 | 37485 | 52628 | 2190 |
| TB + IN-12 + Diverged | 83.61 | 87.17 | 223062 | 516959 | -54051 |
| TB + IN-12 + Class Diverged | 81.06 | 86.33 | 125788 | 205571 | -51354 |

Figure 3: Results for Experiment 1

## 5 EXPERIMENT 2

In Experiment 1, we saw that the network generalizes to unseen images from both datasets almost comparably, irrespective of what kind of perceptual learning signal is used to train the network. This suggests that generalization performance is driven strongly by the category learning signals presented to the network. In the current set of experiments, we seek to investigate whether the network can learn to generalize from inconsistent class pairings between the two datasets. Specifically, we randomly shuffle the labels of the IN-12 images, so that IN-12 cars now correspond to some other category in the Toybox dataset, such as cups. *How well can the network be trained in the case of such inconsistent class labels?* Previous work Zhang et al. (2021) has shown that deep networks can fit random labelings of the training images. However, we are not interested in how well the network memorizes the inconsistent labels. Instead, we are interested in answering the question: *to what extent can the network predict these inconsistent class labels for unseen images from the two datasets?* Further, we seek to understand how the presence(or absence) of perceptual learning signals affects generalization performance in this case. To this end, we vary the perceptual learning signals as shown in Fig 2. The different experimental settings are:

- TB + IN-12R: This is the baseline setting for this set of experiments. In this setting, we randomly shuffle the IN-12 labels and train the network with the inconsistent labels. This setting is similar to the TB + IN-12 setting, except that the IN-12 labels are randomized.

- TB + IN-12R + Aligned: In this setting, the network receives a perceptual signal to align the feature vectors of all Toybox images in a minibatch with those of the IN-12 images.

- TB + IN-12R + Class Aligned: Here, the perceptual signal given to the network tries to align the images associated with a particular label. Note that in this case, since the IN-12 labels have been

randomized, the class alignment signal tries to align the inconsistent class pairings in the feature space. For example, if the shuffling associated IN-12 cars with Toybox cups, the learning signal would try to align Toybox cups with IN-12 cars in the feature space.

The experiments here also have a similar setting as those in the previous section, i.e. we train networks with a ResNet-18 backbone using the Adam optimizer and following similar learning rate schedules as in the previous set of experiments.

**Results**   Fig 4 shows our results for Experiment 2. We see that there exists a key difference between how the network treats inconsistent category vs perceptual signals. In the first variant, despite the inconsistencies in the class labels, the network successfully applies the inconsistent labels to previously unseen images. The same pattern is seen in the second variant (TB + IN-12 + Aligned) too. However, in the third variant, where the network gets the inconsistent perceptual signal, there is a small drop in performance and large drop in the overlap between the two domains, showing that the network effectively ignores the perceptual learning signal.

| Setting | Accuracy | | Avg. Density of Cluster | | Avg. overlap b/w |
| --- | --- | --- | --- | --- | --- |
| | Toybox | IN-12 | Toybox | IN-12 | domains |
| TB + IN-12R | 78.27 | 85.50 | 77315 | 41552 | -30567 |
| TB + IN-12R + Aligned | 77.69 | 84.58 | 12318 | 5552 | -8002 |
| TB + IN-12R + Class Aligned | 76.32 | 82.50 | 10049 | 10629 | -15429 |

Figure 4: Experiment 2 Results

# 6   EXPERIMENT 3

In the previous experiments, we have considered the cases where the network during training had some information about the correspondence between images from the Toybox and the IN-12 dataset. In Experiment 1, the correspondence was consistent, i.e. both Toybox and IN-12 cars were labeled as cars, whereas in experiment 2, it was inconsistent, i.e. Toybox cars and IN-12 ducks were labeled as cars. In experiment 3, we look at the case where there is no known correspondence between the Toybox and IN-12 images, i.e. given an image of a car from the Toybox dataset and another from the IN-12 dataset, the network has no information about whether the ground truth labels for the images are consistent or not. *Are images from corresponding classes in the two datasets near each other in the feature space? Does the network, during training, identify similarities in the semantic information in images from the two domains and put them closer together?*

**Experimental Setup**   The experiments, in this case also follow the joint supervised training framework. However, in this case, we have two classifiers, one for each dataset. During forward propagation, each minibatch contains images from both the datasets. However, after obtaining the features from the ResNet-18 backbone, the minibatch is split — one classifier is fed the images from Toybox and its output is used to calculate the classification loss on the Toybox dataset, while the other classifier is fed the IN-12 images and its output is used to calculate the loss on the IN-12 dataset. Because the classifiers are separate for the two datasets, the network learns to organize the feature space so the Toybox images can be classified effectively and the IN-12 images can be classified effectively, but the network has no information about the relative correspondence between Toybox and IN-12 images. *How is the feature space organized in this case? Does the network organize IN-12 images for a particular class close to Toybox images for that class?* To measure this, after training, we measure the accuracy of the Toybox classifier to classify IN-12 test images. Note that these images are from the test set and were not presented to the network at any point during training. Additionally, the Toybox classifier was never trained using labels for the IN-12 images.

In this setting too, we are interested in the effect of perceptual signals in the final performance. To do this, we do a variant of this experiment where there is an additional loss to perform global alignment between all Toybox images in a minibatch and all IN-12 images in a minibatch. Note that here we are performing global alignment and the network does not know the correspondences between the two sets of classes, only that they belong to the same label set.

**Results**  Table 5 shows our results for experiment 3. Looking at the first row, we see that the Toybox classifier gets 42% accuracy on the IN-12 test images, despite the classifier never having been trained using the IN-12 labels at all. This indicates that the neural network automatically learns some correspondence between the Toybox and IN-12 images for the same class. This is also seen in the overlap value, which achieves a much higher value than previous models where alignment was not part of the training signal. Looking at the second row, we see that the accuracy of the Toybox classifier on the IN-12 images increases further when a global alignment signal is given to the network. Looking at the overlap value, we see that the metric achieves a small positive value, showing some non-trivial overlap between the two datasets.

| Setting | Accuracy | | Avg. Density of Cluster | | Avg. overlap b/w |
| --- | --- | --- | --- | --- | --- |
| | Toybox | IN-12 | Toybox | IN-12 | domains |
| TB + IN-12 2FC | 79.78 | 42.17 | 32060 | 16868 | -1801 |
| TB + IN-12 2FC + Global Aligned | 80.14 | 52.25 | 4289 | 3292 | 142 |

Figure 5: Results for Experiment 3

# 7  RELATED WORK

**Domain Alignment**  Aligning different domains has been a popular approach addressing the distribution shift ML problem. Ben-David et al. (2010) showed that divergence between the two domains together with empirical error on the source domain is a good approximation of the target error; subsequently, many approaches use this idea to address distribution shift. One popular approach has been to use a distance metric based on the Maximum Mean discrepancy (Gretton et al., 2012); it is a distance measure between two distributions defined as the distance between the mean embeddings of the distribution in an RKHS. This metric has been used in several different variants to address problems of domain adaptation (Long et al., 2015; 2017; Ghifary et al., 2014; Kang et al., 2019) and domain generalization (Li et al., 2018).

**Connections between deep learning and cognitive science**  Our work is related to other recent work that leverage recent advances in deep learning to address important questions in the development of visual abilities in human infants. Bambach et al. (2018) demonstrated that CNNs learn better representations when they are trained on the visual experience of infants vs toddlers. In a similar vein, Stojanov et al. (2019) considered the problem of catastrophic forgetting in ML systems and showed that naturalistic patterns of repetition in an infant's visual experience significantly reduce the effect of catastrophic forgetting for visual object recognition. Orhan et al. (2020) looked at the problem of learning representations from videos from infants' play sessions and found that generic self-supervised learning methods can learn powerful high-level visual representations from this data. More recent work (Sanyal et al., 2023) has shown that an infant's natural visual experience presents stronger signals for learning representations than models which have no access to these experiences.

There is also research that takes insights from human visual development to try to build more robust and generalizable models of vision. For example, humans have a shape bias in their visual learning (Landau et al., 1988), and recently, Stojanov et al. (2021) showed that incorporating shape bias into the latent space of a vision system drastically improves generalizability in few-shot learning scenarios. Advances in ML have also helped address important questions about human visual development. Vogelsang et al. (2018) proposed a new theory to explain deficits in configural face processing in children born with congenital cataracts. Other work (Dobs et al., 2019; Jang & Tong, 2021) have used CNNs to investigate face perception and object recognition abilities in humans.

# 8  CONCLUSION

In this work, we investigate howw categorical and perceptual learning signals interact and enable the learning of representations which bridge the distribution gap between the two datasets. We show that the feature space in CNNs can be shaped in different ways without influencing the accuracy on the two datasets. We further show that CNNs learn to apply inconsistent labelings to unseen images from both datasets. Finally, we show that CNNs learn some relationship between corresponding classes from the two datasets even when no such information has been presented through the learning signal.

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
