# OpenReview forum: "Investigating Feature Alignment Under An Infant-Inspired Visual Distribution Shift"
_ICLR.cc/2024/Conference — Submitted to ICLR 2024_

### Official Review · Reviewer_2WNu · 2023-10-27

**Soundness:** 2 fair
**Presentation:** 2 fair
**Contribution:** 2 fair
**Rating:** 3
**Confidence:** 4

**Summary:**

The paper offers a study on aligning object classifications and hidden layer representations between two adapted datasets (toybox and IN12, where the latter integrates images from ImageNet and MS-COCO). It thereby analyses the developing structures in a standard ResNet-18. The work is motivated from cognitive science, where it is known that both categorical learning signals and perceptual learning signals support categorical learning (such as naming an entity the same and having similar feature properties between real and artificial objects, animals, and artifacts ; e.g. toy duck and a real duck).

**Strengths:**

Motivation is appealing.
Paper is reasonably well-structured.
Three experiments provide generally interesting performance effects.
The story and cognitive science motivations sets a nice framework.
The combined dataset seems to have some merit.

**Weaknesses:**

There is nothing really new here – the neural network is very simple. The alignment and classification mechanisms are standard. The experimental variations are interesting. All in all for me a nice workshop paper story.

The story itself is, however, rather far-fetched I am afraid. The perceptual signals here are explicitly aligned – in nature there is not such thing as an explicit alignment between toy and real ducks – or between datasets for that matter. Thus, the alignment story, although certainly interesting, does not really hold in this respect.

The dataset itself has quality issues. The toy dataset shows the hands. The image datasets show (most likely) statistically correlated backgrounds. As a result, the whole investigation is not so much about the respective objects, as the paper indicates -- even if one sees some overlap in the classification performance.

Even when ignoring these weaknesses, I am afraid that the whole story does not provide enough novel insights to warrant publication at a highly competitive conference, such as ICLR. The results are generally expectable. There is no real new insight. The authors also do not really highlight any.

The paper itself could be written in a more approachable manner partially. Section 3 first big paragraphs are hard to understand in the first go. I recommend writing more clearly what you exactly mean by category and perceptual signals (the analogy is far-fetched as indicated above, particularly for the “perceptual signals” case). Section 3.1 is not motivated well at al. Section 4 – the notation is not fully consistent – kernel K and k is the same? (x,y) are sometimes two image instances and sometimes denote image and class of image (in the set notation). It also remains unclear in which layer of the ResNet you apply structural alignment – a small figure would be very useful here. Moreover, Section 4 gives much more detail than experiment 4, while Experiment 2 and 3 and individual Sections.

Results should also report performance on training data and note overfitting. There are not too many classes here, so I am surprised that the accuracy does not even reach 90%.

**Questions:**

Just meant as hopefully helpful recommendations:

Am surprised that you do not cite any work from Linda Smith in the introduction (later there are some), who has done lots of work on active object explorations observed in infants.

Can you motivate the outlier removal better – and possibly illustrate its effect.

I definitely recommend weighting the two loss componenes since L_joint and l_domain have fully different units.

Figure 5 – why not show the results also the other way round (classifying ToyBox with the IN-12 class layer output)? Why are the densities lower than in Figure 3, where the class should, if at all, unify densities and not push them apart (you state at the end of Section 4 that grouping might happen here – would be good to support this with some clustering evidence).

---

> ### Author Response · Authors · 2023-11-23
> **Thank you for your comments and feedback!**
>
> Thank you for the detailed comments and the helpful feedback on our work. They will be extremely beneficial for improving our work. We would like to provide some additional clarifications for some points raised in the review:
>
>
>
> **Quality Issues In Datasets:**
> We disagree that the presence of hands in Toybox and of statistically correlated backgrounds in the instance-rich dataset weaken our results. Hands are pervasive in an infant’s view during active object exploration, while category instances seen in the real world also likely have statistically correlated backgrounds.
>
>
>
> **Structural Alignment:**
> We apply the structural alignment on the outputs of penultimate layer of the network before the classification layer.

---

> > ### Comment · Area_Chair_LALW · 2023-11-23
> >
> > Dear Reviewer,
> >
> > The author has provided responses to your questions and concerns. Could you please read their responses and ask any follow-up questions, if any?
> >
> > Thank you!

---

> > > ### Comment · Reviewer_2WNu · 2023-11-23
> > > **My main concerns were not addressed**
> > >
> > > The authors replied on the last minute and have apparenlty not changed anything in their paper (notice the typo in the conclusion "howw"). They also have not replied to most of my concerns.
> > >
> > > I also have considered the other reviews in my reflection, which are milder in their judgement but generally imply the same.
> > >
> > > Overall, the main concern still fully holds:
> > > "[...] I am afraid that the whole story does not provide enough novel insights to warrant publication at a highly competitive conference, such as ICLR. The results are generally expectable. There is no real new insight. The authors also do not really highlight any. "

---

### Official Review · Reviewer_dPPP · 2023-10-28

**Soundness:** 2 fair
**Presentation:** 2 fair
**Contribution:** 2 fair
**Rating:** 5
**Confidence:** 3

**Summary:**

Motivated by visual learning in people where human infants experience extended amount of experience with small number of objects and a long tail exposure to less familiar objects the paper studies building coherent representations to bridge the distribution shift in images. Paper highlights, the systems learning such representations can leverage two different signals:

1. Categorical learning signals i.e. assigning two different samples same class
2. Perceptual learning signals i.e. based on perceived similarities ex: toy birds and real birds have bills, wings, feet

The paper studies how these two learning signals impact generalization across domains and proposes a new cluster based metric to quantify feature alignment across categories and domains where the metric looks at per-category density, per-category overlap, and full-domain overlap.

Authors find in their first experiments that varying perceptual learning signal doesn’t impact network’s generalization ability to unseen images from image from both distributions. Next, they investigate how network handles inconsistent perceptual and categorical signals and they find that it learns to apply inconsistent categorical labels but ignores inconsistent perceptual labels. Third set of experiments show that network learns relationships between classes from two distributions even when no information is provided.

Overall, this work provides insights into how perceptual and categorical signals enable learning representations to bridge distribution gaps

**Strengths:**

1. The experiment setup and metrics are well designed to systematically probe different configurations of learning signal
2. Experiments in section 6 demonstrate that networks can automatically learn some form of correspondences between images from toybox and IN-12 which enables significantly better than random chance performance. It is also interesting to see that cross-domain performance improves significantly when comparing results from Figure 5 to Rows 1 and 2 from Figure 3.
3. Experiments in section 2 highlight that network is capable of learning inconsitent categorical labels but not capable of learning inconsistent perceptual alignment signals from two distributions.
4. Paper is easy to follow

**Weaknesses:**

1. The dataset used is relatively small and simple with only 12 categories. Additional experiments on larger and more complex datasets would strengthen the claims
2. One key question that is not answered in the paper is - how the ratio of different data distributions affect the generalization ability of the network. Varying size of datasets could impact results significantly, I’d appreciate more discussion on this front
3. The work is motivated by infant visual learning but there is no direct comparison with data collected from infants or human behavioral data across time to back up the cognitive claims.
4. A breakdown of generalization for each category across datasets for all experiment could provide more insights into the results

**Questions:**

1. It would be good if authors can relabel the figures and tables and update the references
2. There’s a typo in the first line of conclusion section “howw” → “how”

---

> ### Author Response · Authors · 2023-11-23
> **Thank you for your review!**
>
> Thank you for the helpful comments and constructive feedback for our work. We would like to provide some additional discussion about some points raised in the review:
>
> **Size of dataset:**
>
> Larger datasets with more categories present one potential direction to further our work. However, one difficulty in this direction is curating datasets with a large sampling of viewpoints for the specific categories. Images from synthetic object datasets could be used to generate data for such larger datasets, but that introduces an additional dimension of distinction between the two domains.
>
> **Effect of ratio of distributions:**
>
> The size of the datasets does play an important role in the generalization ability of the network. In some preliminary experiments, we found that reducing the size of any of the domains leads to a reduction in performance for test images for that domain. We want to explore this further in future work.
>
> **Cognitive motivation:**
>
> Our work is motivated by developmental psychology literature and the two datasets we used contain some properties that have been documented in this literature. Thanks for your comment on better relating to this line of literature in motivating our work and supporting the claims about the important properties of the two datasets.

---

> > ### Comment · Area_Chair_LALW · 2023-11-23
> >
> > Dear Reviewer,
> >
> > The author has provided responses to your questions and concerns. Could you please read their responses and ask any follow-up questions, if any?
> >
> > Thank you!

---

### Official Review · Reviewer_Hbmk · 2023-10-29

**Soundness:** 3 good
**Presentation:** 2 fair
**Contribution:** 3 good
**Rating:** 6
**Confidence:** 5

**Summary:**

This is an interesting study that examines learning through categorical learning signals and perceptual learning signals. The work examines the hypothesis that cross-domain feature alignment can be used as a powerful step towards learning. The paper is nicely conducted and provides useful steps toward solutions to challenges in distribution shifts during learning.

**Strengths:**

The problem examines an important problem on how to learn about objects across large distribution shifts

The experiments are clearly conducted and evaluate important regimes inspired by how animals learn in their environments, with a major distinction between few objects/multiple viewpoints (realistic learning regime) versus multiple object instances / few viewpoints (typical in computer vision and not a realistic learning regime).

**Weaknesses:**

The figures (which are actually tables but called figures throughout the paper) do not have any error bars, let alone any statistics. All the claims from the authors are based on eyeballing the numbers to identify what is or what is not different.

For example, in Figure 3 the authors write that "all the different models achieve comparable accuracy on both the datasets". It is not clear which numbers they are comparing with which numbers here. There are differences in performance. Whether these differences are interesting or not depends on error bars and statistics, which are not presented.

Cluster density for IN12 is stated to be larger than for TB but this seems to depend on the setting in Fig. 3

It is hard to build an intuition for the overlap values. There is a formulate to define overlap on page 5, and the authors comment on the overlap being positive or negative. But it would be useful to get a better intuition for these values. This is easy for accuracy, we know what is the ceiling (100), what is chance (100/numbr of classes), etc. What would be chance overlap? What would be high overlap?

**Questions:**

The main suggestion is to add error bars and statistics throughout to document whether the changes reported are meaningful or not (and also to provide rigorous quantification in those cases where the authors state that there are no changes).

The overall writing is very sparse, both the results within each section and especially the conclusion section.

Providing a better intuition for the numbers reported under density and overlap would help interpret those metrics. Especially useful are comparisons with chance values, and ceiling values. Additional useful metrics could include experiments for those metrics under different manipulations (e.g. classes composed of mixtures of labels, within class variability, etc).

---

> ### Author Response · Authors · 2023-11-23
> **Thank your for the review!**
>
> Thank you for your comments and feedback on our submission. They will be helpful for improving our work.

---

### Official Review · Reviewer_dHsC · 2023-10-31

**Soundness:** 2 fair
**Presentation:** 2 fair
**Contribution:** 2 fair
**Rating:** 5
**Confidence:** 2

**Summary:**

The paper curated a novel dataset with the same set of categories as a dataset mimicking visual experience of infants but includes limited viewpoints of each object and larger numbers of objects in each category. The newly curated dataset and an existing dataset mimicking infants' visual experience form a domain shift.

Using new metrics proposed by the paper, the work investigated how categorical and perceptual learning signals impacts CNN's ability to generalize across domains and the density of cluster and overlap of the learned features between the two domains.

**Strengths:**

The proposed pair of datasets is a good approach to investigate domain generalization with distribution similar to what infants experience: frequent experience of few examples with sparse experiences of many other examples.

The paper included several well-thought experiments that either encourage or discourage alignment of embedding at different levels of granularity.

It found a few interesting phenomenon:
When the categorical learning signals are fixed (the network has access to correct labels) but different loss are used to bias representation alignment, the performance seems not impacted.
When the labels of the two datasets are shuffles, the performance of the learned classifiers still remain high on both datasets with or without additional losses attempting to align the representation between the two datasets, except for a minor drop when the additional losses encourage alignment at categorical level (with shuffled labels)
Finally, when a CNN backbone is shared between datasets but two classifiers are trained on top of the CNN, there is still an above-chance generalization performance from the dataset of frequent experience of individual objects to the dataset of infrequent experiences of many objects.

**Weaknesses:**

Although the evaluation on classification accuracy is sound, I have strong reservation against performing all the analysis of cluster density and overlapping between domains on a UMAP-preprocessed version of the embedding. While UMAP attempts to preserve the representational structure in a low-dimensional space, ultimately it is primarily a visualization technique. Since the real behavior of the network should be on the original feature space, I am not sure what UMAP brings to the table in this analysis. Wouldn't the entire analysis on the original embedding illustrate better how the overlap and density of clusters differ across experiments? Not that I prefer seeing a UMAP illustration, but using a dimension reduction technique that allows visualization but do not actually use its power to visualize anything interesting seems strange as well.

I also feel that more illustration of what perceptual signals mean by the author and what they capture in the experiments will be very helpful. The introduction explains perceptual learning signals as "implicitly assign two inputs to the same class because of perceived similarities". But I think the "perceived similarity" is not something fixed in the nature but strongly depends on the way a representational system selects features. In other words, it is a consequence of learning (with some objective), rather than a teaching signal, unless we trust a randomly initialized CNN already gives us a good signal for what objects are "perceptually signal". Using the example in the paper: it is true that both toy ducks and real ducks have bills, wings and webbed feet, but on the other hand, they also have clear distinction in whether they have feather, whether their body are of pure color (toy ducks often do while real ducks often do not). If we have not had the preference of using the features listed by the authors to decide category boundary but have instead relied on the features I mentioned, then we would not put them into the same category. The preference of features being used for classification by the brain is likely a consequence of learning. The manipulation in the experiment that is called "perceptual signals" are mainly about aligning representation across datasets, which does not seem to manipulate the preservation or removal of either the types of features listed by the paper for the duck example or the ones I listed.

Another way to think of the point 1 and 2 in the abstract is that one can think that the preference of features for deciding perceptual similarity is a consequence of being forced to find commonality in objects with the same labels in two datasets. If this is the case, then the perceptual learning signal seems to only play a role in the context of meta-learning or continual learning when learning new categories after learning some other categories, which is not reflected in the experiments in this paper.

In the minimal case, if any claim about perceptual signals is to be made, I feel that some visualization of the features being used in classifiers learned in different experiments that show significant difference in performance or the metrics proposed by the paper would be very informative for understanding what these perceptual signals are.

**Questions:**

My suggestion is consistent with my comment of weakness:
I think a direct evaluation of the metrics on the original embedding is necessary and interesting.
A better visualization of what the authors mean by perceptual (learning) signals would be very important for understanding the conclusions of the paper. A better link between the definition of perceptual learning signals and the experiments should ideally be established or explained more clearly.

---

> ### Author Response · Authors · 2023-11-23
> **Thank you for your review!**
>
> Thank you for your detailed comments and helpful feedback about our work. They will be beneficial for improving our work. We would like to provide some additional details and clarifications pertaining to some questions raised:
>
> **Using UMAP and metrics for high-dimensional feature spaces:**
>
> The feature space of the trained neural network is high-dimensional and interpreting high-dimensional spaces is complicated. For this work, we were interested in exploring the feature space of the trained neural networks. Previous work has shown that the UMAP can be used to find interesting structures in the data, hence our reliance on UMAP for the data analysis.
>
> Having said that, we agree that evaluating the metrics on the original embeddings is important because the embeddings drive the behavior of the neural network.
>
> **Perceptual Signals:**
>
> Yes, it is interesting to consider how the perceptual learning signal can vary depending on context. A toy duck and a real duck are similar in that they are both ducks, but they are different in that one is a toy. It would be interesting to consider how these context-sensitive signals can be used to drive learning.
>
> For this submission, we assume that perceptual learning signals indicate the similarity or dissimilarity between two groups of images for the same category. For example, in the aligned case, we use a maximum mean discrepancy (MMD) based loss function to bring the images from the two datasets closer in the feature space. In the diverged case, we use an additional task of predicting the dataset for each image as an auxiliary task to encourage the network to separate the features for images from the two datasets.

---

> > ### Comment · Area_Chair_LALW · 2023-11-23
> >
> > Dear Reviewer,
> >
> > The author has provided responses to your questions and concerns. Could you please read their responses and ask any follow-up questions, if any?
> >
> > Thank you!

---

### Official Review · Reviewer_5jV7 · 2023-11-03

**Soundness:** 3 good
**Presentation:** 3 good
**Contribution:** 3 good
**Rating:** 6
**Confidence:** 3

**Summary:**

This paper investigates a novel learning scenario: transferring knowledge learned by infants by manipulation of small number of objects in numerous poses to much larger collections of objects in canonical poses. They do a series of experiments investigating under what circumstances the representations and classification learned in one scenario transfer to the other.

**Strengths:**

The authors pose a very interesting question relevant to both child development and to machine learning: how can we extrapolate from a small set of highly labelled data to data in-the-wild? As someone not familiar with this literature, I found the motivation and exposition lucid. The experiments start to answer questions about how this kind of domain generalization can occur.

**Weaknesses:**

I thought the experiments were a little thin. These are, overall, a very small set of training scenarios (~10). There are a lot of interesting questions to explore, which I don't expect the authors to fully cover throughout this paper. However, I do think a big blind spot of the paper is the lack of consideration for unsupervised or self-supervised learning. In an infant learning scenario, we know that the supervisory signals are very sparse. It has long been suspected that self-supervision/unsupervised learning is more plausible for learning visual representations, and self-supervised representations align well with those of humans (e.g. Zhuang et al. 2021). In that context, I would like to see how self-supervised learning leads/doesn't lead to aligned representations across these two scenarios.

The paper has a lot of small experiments which differ slightly from one another, and I found myself moving back and forth between tables and the definitions of the different experiments. The authors should make the figure captions more informative than "Results for Experiment N".

**Questions:**

-

---

> ### Author Response · Authors · 2023-11-23
> **Thanks for your comment!**
>
> Thank you for your comments and the helpful feedback on our submission. They will be helpful in improving our work.
>
> **Q: Unsupervised Learning**
>
> Yes, we agree that supervisory signals are sparse in infants’ learning environments and that it is likely that infants self-supervise their learning processes. We also agree that there are many important and interesting questions pertaining to this area. However, that is not the focus of this paper. In this submission, we ask how different kinds of learning signals lead to different kinds of learning outcomes as measured by classification accuracy and alignment between the different datasets. We assume the presence of perfect supervisory knowledge, as given by class labels for each image. Even in this case, the effect of different learning signals on the learning outcomes is not well understood. In this paper, we focus on this aspect.

---

> > ### Comment · Area_Chair_LALW · 2023-11-23
> >
> > Dear Reviewer,
> >
> > The author has provided responses to your questions and concerns. Could you please read their responses and ask any follow-up questions, if any?
> >
> > Thank you!

---

### Meta-Review · Area_Chair_LALW · 2023-12-06

**Metareview:**

The paper introduces categorical learning and perceptual learning. Via carefully crafted experiments, the paper studies the effect of their interactions on cross-domain classification problems. To quantify feature alignment, a cluster-based metric is proposed.

5 reviewers gave their initial feedback during the pre-rebuttal period. All of them also interacted with the authors at least once during the rebuttal period. All of them kept their initial ratings the same after the rebuttal, as their concerns were only partially addressed or not addressed at all. Given the current status of the paper, AC recommends rejection.

Strengths and weaknesses are summarized below:

Strengths:
1. The problem setting, inspired by infant learning, is very interesting.
2. The paper is well written though clarity in certain parts is still required

Weaknesses:
1. The clustering and metrics should be calculated in the original embedding space.
2. The idea of seeking inspiration from infants' learning is interesting; however, the claim seems to be hand-wavy. For example, there is no benchmarking with real infant data, such as behavioral data. There is no account for self-supervised learning, either.
3. Several reviewers raise concerns/questions about the metric designs and interpretations. Their definitions and motivations can be made clearer.
4. The experiments tend to be limited to small-scale datasets with a few number of object categories. Results are missing statistical tests, which are necessary to verify the conclusions made by the paper.

AC strongly recommends the authors revise based on the feedback from the reviewers and resubmit the paper to future venues.

**Justification For Why Not Higher Score:**

see weakness above

**Justification For Why Not Lower Score:**

NA

---

### Decision · Program_Chairs · 2024-01-16

Reject